# Dysglycemia and Abnormal Adiposity Drivers of Cardiometabolic-Based Chronic Disease in the Czech Population: Biological, Behavioral, and Cultural/Social Determinants of Health

**DOI:** 10.3390/nu13072338

**Published:** 2021-07-08

**Authors:** Iuliia Pavlovska, Anna Polcrova, Jeffrey I. Mechanick, Jan Brož, Maria M. Infante-Garcia, Ramfis Nieto-Martínez, Geraldo A. Maranhao Neto, Sarka Kunzova, Maria Skladana, Jan S. Novotny, Hynek Pikhart, Jana Urbanová, Gorazd B. Stokin, Jose R. Medina-Inojosa, Robert Vysoky, Juan P. González-Rivas

**Affiliations:** 1International Clinical Research Centre (ICRC), St Anne’s University Hospital Brno (FNUSA), 65691 Brno, Czech Republic; anna.polcrova@fnusa.cz (A.P.); maria.garcia@fnusa.cz (M.M.I.-G.); geraldo.neto@fnusa.cz (G.A.M.N.); sarka.kunzova@fnusa.cz (S.K.); maria.skladana@fnusa.cz (M.S.); jan.sebastian.novotny@fnusa.cz (J.S.N.); gorazd.stokin@fnusa.cz (G.B.S.); juan.gonzalez@fnusa.cz (J.P.G.-R.); 2Department of Public Health, Faculty of Medicine, Masaryk University, 62500 Brno, Czech Republic; vysoky.rob@gmail.com; 3Research Centre for Toxic Compounds in the Environment (RECETOX), Masaryk University, 62500 Brno, Czech Republic; hynek.pikhart@recetox.muni.cz; 4The Marie-Josée and Henry R. Kravis Center for Cardiovascular Health at Mount Sinai Heart, Icahn School of Medicine at Mount Sinai, New York, NY 10029, USA; jeffreymechanick@gmail.com; 5Division of Endocrinology, Diabetes and Bone Disease, Icahn School of Medicine at Mount Sinai, New York, NY 10029, USA; 6Department of Internal Medicine, Charles University Second Faculty of Medicine, 10506 Prague, Czech Republic; zorb@seznam.cz; 7Foundation for Clinic, Public Health, and Epidemiology Research of Venezuela (FISPEVEN INC), Barquisimeto 3001, Lara, Venezuela; nieto.ramfis@gmail.com; 8Department of Global Health and Population, Harvard TH Chan School of Public Health, Harvard University, Boston, MA 02115, USA; 9LifeDoc Health, Memphis, TN 38119, USA; 10Second Department of Internal Medicine, St. Anne’s University Hospital in Brno and Faculty of Medicine, Masaryk University, 65691 Brno, Czech Republic; 11Department of Epidemiology and Public Health, University College London, London WC1E6BT, UK; 12Center for Research in Diabetes, Metabolism and Nutrition, Second Department of Internal Medicine, Third Faculty of Medicine, University Hospital Královské Vinohrady, 10000 Prague, Czech Republic; urbja@seznam.cz; 13Division of Preventive Cardiology, Department of Cardiovascular Medicine, Mayo Clinic, Rochester, MN 55905, USA; MedinaInojosa.Jose@mayo.edu; 14Marriott Heart Disease Research Program, Mayo Clinic, Rochester, MN 55905, USA; 15Department of Health Support, Faculty of Sport Studies, Masaryk University, 62500 Brno, Czech Republic

**Keywords:** adiposity, cardiometabolic risk, cardiovascular disease, chronic disease, dysglycemia, insulin resistance, nutrition, obesity, type 2 diabetes

## Abstract

In contrast to the decreasing burden related to cardiovascular disease (CVD), the burden related to dysglycemia and adiposity complications is increasing in Czechia, and local drivers must be identified. A comprehensive literature review was performed to evaluate biological, behavioral, and environmental drivers of dysglycemia and abnormal adiposity in Czechia. Additionally, the structure of the Czech healthcare system was described. The prevalence of obesity in men and diabetes in both sexes has been increasing over the past 30 years. Possible reasons include the Eastern European eating pattern, high prevalence of physical inactivity and health illiteracy, education, and income-related health inequalities. Despite the advanced healthcare system based on the compulsory insurance model with free-for-service healthcare and a wide range of health-promoting initiatives, more effective strategies to tackle the adiposity/dysglycemia are needed. In conclusion, the disease burden related to dysglycemia and adiposity in Czechia remains high but is not translated into greater CVD. This discordant relationship likely depends more on other factors, such as improvements in dyslipidemia and hypertension control. A reconceptualization of abnormal adiposity and dysglycemia into a more actionable cardiometabolic-based chronic disease model is needed to improve the approach to these conditions. This review can serve as a platform to investigate causal mechanisms and secure effective management of cardiometabolic-based chronic disease.

## 1. Introduction

In contrast to the decreasing burden and mortality related to cardiovascular disease (CVD) and certain related CVD risk factors, such as hypertension, dyslipidemia, and tobacco use, the two key mechanistic drivers of dysglycemia and abnormal adiposity are increasing in prevalence in Czechia. From 1990 to 2019, the number of disability-adjusted life years (DALYs) increased by 18.6% for dysglycemia and by 10.4% for body mass index (BMI) > 23.0 kg/m [1,2]. In 2020, epidemiological data affirmed that both dysglycemia and abnormal adiposity increased the risk of severe COVID-19 [2], highlighting the need for urgent implementation of cardiometabolic risk mitigation strategies [3]. On a global scale, there is an array of both biological and cultural/social determinants of disease that interact and drive cardiometabolic-based chronic disease (CMBCD) [4]. However, the specific nature of each driver and how they interact to impel progression of CMBCD can vary.

This review evaluates the key drivers for dysglycemia and abnormal adiposity in Czechia based on the epidemiology of various biological and cultural/social determinants of health. Particular variables of interest include demographic characteristics, modifiable cardiometabolic risk factors (dysglycemia, abnormal adiposity, hypertension, dyslipidemia, healthy eating, physical activity, and tobacco use), and ethno-cultural and social-economic factors (healthcare infrastructure, health literacy, disparities in access to quality healthcare, housing/income/educational levels, and mental health). By analyzing the results of a formal literature searching protocol on cardiometabolic drivers in Czechia, discrepancies, emergent relationships, and essential elements of an effective preventive care plan can be identified. Furthermore, this methodology can be replicated for other regions of the world to expose epidemiological differences that warrant refinements and adaptations to current, generalized CMBCD models.

## 2. Materials and Methods

This narrative review analyzes the epidemiology of cardiometabolic risk factors as well as key drivers of abnormal adiposity and dysglycemia, including biological (genetic), behavioral (physical activity, eating patterns, smoking, alcohol consumption, mental health), and environmental (health literacy, health inequality) drivers. To summarize the evidence, a comprehensive literature review in MEDLINE (PubMed), Web of Science, and Scopus databases, spanning 1980 to 2020 inclusive, was performed by two researchers (I.P. and A.P.) in December 2020–January 2021 using the following keywords: “Czechia” OR “Czech Republic” OR “East* Europ*”, and terms related to cardiometabolic risk factors: “prevalence” AND “hyperten*”, OR “obes*”, OR “diabetes”, OR “prediabetes”, OR “blood glucose”, OR “hyperlipid*”, OR “dyslipid*” OR “cardiovascular disea*” OR “metab* syndrome”. Terms related to behavioral drivers were “diet*”, OR “physical* activ*”, OR “exercise”, OR “alcohol”, OR “body image”, OR “depress*”, OR “anxiet*”. Terms related to environmental drivers were “health literacy”, OR “health inequalit*”. Terms related to biological drivers were “gen* varian*”, OR ”gene”, OR “”gen* polymorph*”, OR “SNP”. For article searching in MEDLINE (PubMed), the following Medical Subject Headings (MeSH) terminology was used: (cardiovascular disease) OR (abdominal obesity) OR (central obesity) OR (obesity) OR (high blood pressure) OR (dyslipidemia) OR (hyperlipidemia) OR (hyperglycemia) OR (prediabetes) OR (type 2 diabetes mellitus) OR (diet) OR (exercise) OR (alcohol drinking) OR (mental health) OR (Body Image) OR (Body Dissatisfaction) OR (Weight Prejudice) OR (Health Literacy) OR (Healthcare Disparities) OR (genes) OR (Polymorphism, Genetic) OR (Europe, Eastern) OR (Czech Republic). The inclusion criteria were observational (retrospective, cross-sectional, and prospective) studies on adults 18 years or older in Czechia. Exclusion Criteria were clinical trials, case studies, studies that were not performed on humans, and studies that included children and adolescents.

Additionally, we ran searches and reported on the sociodemographic characteristics of Czechia and the structure of healthcare system. Data from the World Health Organisation, Global Burden Disease, Czech Statistical Office, Institute of Health Information and Statistics of the Czech Republic, European Commission, Ministry of the Interior of the Czech Republic, Czech Public Insurance Company, Czech Diabetes Society, Czech Obesity Society, The World Bank, web-pages of various health-promoting Czech projects, and others were included to describe specific categories of risk factors.

### Variables Definitions

Current smoking was reported as smoking daily or less than daily. Overweight was reported as body mass index (BMI, kg/m^2^) between 25.0 and 29.9, and obesity as BMI ≥ 30. Hypertension was reported as systolic blood pressure (SBP) ≥ 140 mmHg and/or a diastolic blood pressure (DBP) ≥ 90 mmHg, or current treatment with antihypertensive drugs. Hypertension control was reported as achieving SBP < 140 mmHg and DBP < 90 mmHg with drug treatment. Prediabetes was reported as fasting blood glucose between 5.6 to 6.9 mmol/L, without treatment or personal history of diabetes. Diabetes was reported as a personal history of diabetes or treatment, or fasting blood glucose ≥7 mmol/L. The criteria to define dyslipidemia and physical inactivity varied across the studies and are reported according to the studies in the results section. Adiposity-based chronic disease (ABCD) [5] was reported in relation to the presence of an abnormal amount of adiposity (defined as BMI ≥ 25.0 kg/m^2^ or a high percent of fat mass), distribution (defined as an increased waist circumference). Subjects with ABCD were categorized as stage 0—those with no identifiable adiposity-based complications (also referred to as “metabolically healthy”); stage 1—those with mild to moderate adiposity-based complications; and stage 2—those with severe adiposity-based complications [5]. Dysglycemia-based chronic disease (DBCD) [6] was defined as stage 1—“insulin resistance” (inferred risk from abdominal obesity and family history of diabetes); stage 2—“prediabetes”; stage 3—“T2D”; and stage 4—“vascular complications” (T2D with personal history of CVD) [6]. 

## 3. Results

A total of 7255 articles were retrieved from the databases, and 1809 duplicates were excluded. Among the 5446 unique articles, 5114 were excluded after reading their titles and/or abstracts. Thus, 332 articles were subjected to a full-text review: we excluded 23 studies analyzing the data from the same epidemiological studies, and another 131 studies were not aligned with the purpose of the review. Finally, 38 studies were included in our narrative review (10 on biological risk factors, 11 on cardiometabolic risk factors, 11 on behavioral risk factors, 3 on environmental risk factors, and 3 on mental health). Additionally, a hand search using references in identified articles was performed.

### 3.1. Epidemiology of Cardiometabolic Risk Factors

Biological characteristics (genetic predisposition) are a powerful driver for the presence of dysglycemia and abnormal adiposity across diverse populations. A summary of important biological determinants for dysglycemia and abnormal adiposity in the Czech population is presented in Table 1.

Cardiovascular disease mortality decreased in Czechia from 1989 to 2019, and this downward trend can be attributed to a decreasing prevalence of CMBCD risk factors and lower impact of CMBCD drivers [17] (Table 2). Among the major risk factors with declining rate are the prevalence of tobacco use in men (from 45% in 1989 to 23.9% in 2016), hypertension in women (from 42.5% in 1989 to 33.5% in 2016), and dyslipidemia in both sexes (men: from 87.7% in 1989 to 74.8% in 2016; women: from 87.5% in 1989 to 69.9% in 2016). The use of lipid-lowering medications increased more than two-fold in both sexes and reached 14.6% in men and 10.0% in women in 2016 (with statins comprising 78.8% of all the prescribed lipid-lowering drugs). The number of individuals treated by antihypertensive drugs has increased in both genders, and a rate of hypertension control improved from 3.9% in 1985 to 32.9% in 2017 [17].

The prevalence of obesity in men has shown an increasing trend over the past 30 years (from 19.7% in 1985 to 37.7% in 2016) [17], while in women, there was no change (28.0% in 1985 to 27.6% in 2016) [17] (Table 2). The prevalence of diabetes has shown an increasing trend in both sexes from 6.9% and 5.4% in 1985 [17] to 11.5 % and 8.3 % in 2014 [18] in men and women, respectively.

Using a new chronic care model, the prevalence of ABCD [5] in the Czech population was 62.8% [25]. Only 2.3% of those were metabolically healthy (Stage 0 ABCD), 31.47% with mild adiposity-related complications (stage 1 ABCD) and 29.1% with moderate and severe adiposity-related complications (stage 2 ABCD). Total ABCD as well as all of the ABCD stages, except for stage 1, were more prevalent in men. Using another new chronic care model, type 2 diabetes (T2D) was interpreted as dysglycemia-based chronic disease (DBCD) based on a spectrum ranging from insulin resistance (present in 54.2 % of the population) to prediabetes (in 10.3 % of the population), to T2D (in 3.7% of the population), and to vascular disease (in 1.2% of the population) [6] (Table 3). DBCD stages 2, 3, and 4 were more prevalent in men; however, total DBCD prevalence was higher in women.

### 3.2. Sociodemographic Characteristics

Czechia is an unitary state with a representative democracy, a parliamentary republic, and decentralized administration with 14 regions and over 6000 municipalities [27]. The Czechian territory covers an area of 78,866 km^2^ and was established in 1993 by the division of Czechoslovakia into Czechia and Slovakia. Since 2004, Czechia has been a member of the European Union [27]. Czechia is located in Central Europe and, according to the World Bank, is categorized as a high-income country [28] with a gross domestic product of EUR 20,900 per capita in 2019. The main industries include engineering, food processing, chemical, and metallurgy [27]. The unemployment rate was 2.0% in 2019 [29]. Czechia had a population of 10,699,142 in 2020 [30]. Czechs belong to the Europid race, with the ethnic minority Roma representing 2.2% of the total population, and foreigners (mainly Ukrainians, Slovaks, Vietnamese, and Russians) around 5.3% [31]. The Czechian population is linguistically homogenous; the official language is Czech [27]. About 94% of adults aged 25–64 years have at least upper secondary education [27].

Czechia is denominationally neutral (no official religion) and freedom of religion is granted [27]. According to the census of 2011, about 14% of the population belonged to a religious denomination, while another 7% declared themselves as believers [32]. In 2019, there were 41 churches and religious societies, the largest being the Roman Catholic Church, Czech Brethren Evangelic Church, and the Czechoslovak Hussite Church [32]. Life expectancy at birth (years) increased from 67.5 in men and 73.3 in women in 1960 to 76.2 in men and 82.1 in women in 2018. The total fertility rate decreased from 2.1 to 1.7 births per woman from 1960 to 2018 [33]. These statistics reflect a demographic population aging, with the proportion of individuals older than 65 years (19.6%) surpassing those younger than 15 years (15.9%) [33]. These descriptors of the Czechian population provide context for interpretation of the following results.

### 3.3. Eating Patterns 

The Eastern European and Czech eating patterns are characterized by high consumption of saturated fats, salt, and alcohol, with insufficient intake of fresh fruit, vegetables, whole grains, and fish [34]. Highly processed foods (commercially prepared, which require no or minimal domestic preparation) comprised more than 70% of the mean energy intake in the region [35]. Consumption of processed meat is a traditional part of the Eastern European diet [34]. Despite of the relatively small population, in 2018, Czechia placed 14th place in regard to the production of processed meat in the world, with 14,000 tons produced annually [36]. Epidemiological studies assessing eating patterns showed that 46.3% of the Czech population did not consume fruits and vegetables daily, and only about 10% consumed at least five portions of fruits and vegetables per day [37]. There was a high weekly consumption of poultry, pork, salami, and sausage, although beef and fish were consumed less than once a week [38,39]. The most frequently consumed cereal was wheat in the form of flour, bread, and other bakery products. Rye and rice were consumed in smaller quantities [39,40]. Being male, having a low income, being physically inactive, and smoking tobacco were associated with a lower intake of fresh produce, legumes, whole grains, nuts, seeds, and milk [41]. Men, as well as individuals with low-to-medium education levels, were also more likely to eat red and processed meat [41]. 

Since 1989, there has been an increase in the consumption of healthy foods: pulses (from 1.3 to 2.8 kg/person/year), nuts (from 2.6 to 3.6 kg/person/year), vegetable oils (from 12.5 to 17.2 kg/person/year), vegetables (from 68.7 to 87.3 kg/person/year), and fruits (from 70.5 to 80.4 kg/person/year). On the other hand, there was a reported decrease in the consumption of milk and dairy products (from 259.6 to 247.5 kg/person/year), meat (from 97.4 to 80.3 kg/person/year), lard (from 6.8 to 4.5 kg/person/year), butter (from 9.4 to 5.4 kg/person/year), refined sugar (from 39.8 to 34.1 kg/person/year), and salt (from 6.3 to 5.7 kg/person/year) [42].

Traditional Czech dishes are rich in red meat, potatoes, gravies, and root vegetables. Common food preparation techniques include stewing, roasting, simmering, deep frying, and hot smoking. Some of the most common traditional Czech dishes are “vepřo knedlo zelo” (roasted pork with bread dumplings and a side of braised cabbage), “sekaná” (Czech version of meatloaf, made of minced pork and beef meat, bacon, onions, and garlic), and guláš (meat stew, usually served with bread dumplings or slices of dark bread). However, there are also vegetarian alternatives, such as “koprovka” (dill creamy sauce, served with boiled eggs and dumplings), “čočka na kyselo” (lentils, served with boiled eggs and pickled cucumbers), and “ovocné knedlíky” (yeast/curd cheese/potato dough dumplings filled with seasonal fruits, sprinkled with sugar, grated curd cheese, and melted butter). Soups, including vegetable soups, broth, and legume soups, are also an important part of Czech cuisine.

According to the World Health Organization (WHO), consumption of pure alcohol in Czechia was 14.4 L per person per year, in 2016. This is 47% higher than the average amount in the WHO European region, and one of the highest globally [43]. On average, one Czechian inhabitant consumes 144 L of beer, 19 L of wine, and 7 L of distillates per year, with 6.1–9.5% consuming alcohol daily [44]. In a population-based, cross-sectional survey in Czechia [45] on alcohol consumption in the past 12 months, 28.2% of men and women did not drink alcohol; 45.6% consumed less than 7 or 14 standard drinks per week for women and men, respectively; and 26.2% had at least 7 or 14 standard drinks per week for women and men, respectively.

### 3.4. Physical Activity

The prevalence of physically inactive adults (performing less than 150 min of moderate-intensity and less than 75 min of vigorous-intensity physical activity per week) in Czechia has increased from 31.4% in 2013 to 42.7% in 2017 [46]. The time spent sitting per day had also increased from 58% in 2005 to 62% in 2017 [47]. Currently, only 60% of Czechs walk for ten min or more at least four days per week [47]. Additionally, less than 16% engage in moderate and vigorous activities more than three days a week [47]. Age and sex inequalities are present in the physical activity patterns of the Czech population. Subjects older than 65 years and middle-age adults (40–64 years old) have a 4 and 1.7 times higher chance, respectively, to be physically inactive in comparison with young adults (18–39 years old) [48]. Moreover, men have a 1.4 times higher chance to be physically active as compared to women [48].

The most popular team sports in Czechia are football, floorball, ice hockey, volleyball, basketball, and kickball (an original Czech sport in which the players kick the ball over a low net). The most popular individual activity is walking, followed by jogging, running, cycling, hiking, inline and ice skating, and skiing [49]. Dog walking is also a frequent activity, as Czechia has the second-highest rate of dog ownership in Europe (38%) [50]. Local evidence suggests that dog owners display a better cardiovascular health profile than that of non-owners [51].

### 3.5. Health Literacy 

Health literacy reflects one’s knowledge, motivation, and competency to access, understand, appraise, and apply health information in order to make judgments and decisions in everyday life [52,53]. Self-efficacy is the confidence of individuals’ own ability to perform certain tasks and to attain determined goals; it is closely linked with the level of health literacy and is associated with better cardiometabolic outcomes [54]. In 2014, in Czechia, the prevalence of low level of health literacy was 59% (below the European average 47.6%) [55], higher in subjects with older age (≥40) and with level of education lower than university degree [55]. In diverse populations, a low level of health literacy is associated with increased risk of T2D [56], less knowledge about T2D [57], lower adherence with diabetes medications [58], and higher risk for T2D related-complications [59]. There was no evidence found that assessed the association between health literacy and dysglycemia/adiposity drivers in the Czech population, nor were there interventions to specifically improve health literacy in these patients. 

### 3.6. Health Inequalities 

In Czechia, health inequalities have been reported across three dimensions [60]:Education. Compared with those with higher education, less-educated men and women showed 11 and 3 years of lower life expectancy, respectively [60]. In a cohort study assessing 8449 adults aged 45–69 years followed for 12 years, education below the upper-secondary level was the strongest determinant of cardiovascular mortality, followed by hypertension and smoking [23]. Additionally, lower education levels have been associated with a higher presence of prediabetes and diabetes [18], as well as the presence of adiposity and dysglycemia-related complications, compared with those with higher educational levels [25,26].Income. In a population-based study, subjects with a lower household income (<EUR 1200 per month) were more likely to present with dysglycemia and adiposity-related complications than those with a higher income (>EUR 1800 per month) [25,26]. Of note, 80% of Czech subjects with a higher income considered their health to be good compared to only 42% of subjects with a lower income [60].Access to healthcare. There was a low level of disparity in access to quality healthcare and specialized health services among different income and education groups [60]. Nevertheless, in certain rural regions, especially Liberecký, Ustecký, Zlinský, and Středočěský, there was a lack of primary care clinicians [60,61].

### 3.7. Mental Health

A few studies, including small samples in Czechia, found that individuals with dysglycemia or abnormal adiposity experience more stress and depressive symptoms [62,63,64], anxiety [63], and lower quality of life [63]. Nevertheless, population-based studies evaluating these associations are lacking in Czechia. 

### 3.8. Healthcare System

The healthcare system in Czechia is based on an obligatory participation of insured persons, with free-for-service healthcare funded by mandatory employment-related insurance plans since 1992 [65]. Health insurance fully covers preventive, diagnostic, ambulant and hospital care (excluding cosmetic surgeries without underlying health reasons), spa care, and patient transport. Insurance also fully or partially covers dental care, medicines and medicinal aids [66]. If that total amount of supplementary payments paid for partially covered medicines exceeds CZK 5000 (EUR 197) per year, the health insurance is obliged to pay the exceeding amount back to the person [67].

#### 3.8.1. Diabetes Care

In Czechia, general practitioners provide primary diabetes care for approximately 27.4% of patients with diabetes [68] and are the healthcare professionals who diagnose the disease most frequently [68,69]. A series of regular preventive healthcare checks is established and is covered by insurance [70]. Patients with poor metabolic control, presence of complications, or in need of multi-drug treatment, including insulin, are usually referred to a diabetologist, but overall care is still shared between a general practitioner and diabetologist [71]. Lifestyle and rehabilitation programs are provided by physicians, as well as physiotherapists and dietitians, with close collaboration.

For patients with diabetes, 25 centers of specialized care (“Centers of Diabetology”) are available, offering initial consultations and ongoing care for patients with insulin pumps, severe diabetes complications, or organ failure caused by diabetes. The diabetes transplant program is provided by one of these centers [72]. The diabetes care system in Czechia also offers specialized centers of foot care (33 centers), education (41 centers), and psychological care (42 centers) for patients with diabetes [72]. In short, comprehensive diabetes care is generally covered by health insurance in Czechia [69,73,74].

According to a Czech cross-sectional study focusing on patients with T2D, results showed that 34.2% of patients achieved an HbA1c <7% (53 mmol/mol), regardless of treatment modality [75]. The diabetes control in subjects using insulin for the treatment study (DIAINFORM) showed that the frequency of acceptable metabolic control (HbA1c <7% (53 mmol/mol)) in patients with T2D treated with insulin was 37.1% [76]. 

#### 3.8.2. Obesity Care

In Czechia, obesity care is multidisciplinary, provided by general practitioners, physicians specializing in diabetology/endocrinology, dietitians, physiotherapists, and psychologists [77]. The level of care depends on the stages of adiposity-related complications. For patients with obesity, there are 19 specialized obesity clinics and 5 centers offering an individualized approach to prevent and treat obesity, and there are 8 centers providing metabolic and bariatric surgery [78].

#### 3.8.3. Health Promotion Policies

A summary of the most successful health-promoting projects implemented in Czechia is presented in Table 4. The large scale of those initiatives and the positive impact of the dysglycemia and abnormal adiposity drivers face many challenges, including lack of a scientific understanding of these drivers among diverse population groups, such as implementation, application of project management, use of organizational and procedural tools, and improving cooperation among stakeholders [79].

## 4. Discussion

This comprehensive review addresses the biological, cultural, sociodemographic, and healthcare-related infrastructural factors influencing the burden of dysglycemia and abnormal adiposity in the Czech population within the context of CMBCD risk mitigation. In the last three decades, the prevalence and mortality rates related to dysglycemia-related complications have increased, while mortality rates related to adiposity-related complications are roughly the same. These prevalence rates are driven by an aging population, an unhealthy Easter European eating pattern, with traditionally high consumption of alcohol above the European average levels, physical inactivity, and time spent sitting per day. These drivers heterogeneously affect the population, with a greater impact on subjects with lower education, income, and rural housing. A low level of health literacy has been reported, which is an important health determinant related to a higher risk of mortality and morbidity worldwide. Overall, these social determinants affect population health despite the huge effort of an advanced healthcare system built on an insurance model that covers the entire population coupled with the availability of multiple specialized care centers for those with obesity and T2D. Unfortunately, these centers typically treat patients with advances stages of the diseases, leaving a wide practice gap for prevention at much earlier stages of ABCD, DBCD, and CMBCD. There is not enough evidence regarding association of abnormal adiposity and dysglycemia with mental health. Worldwide, obesity and T2D are related to perceived stigma [88] and have been associated with an increased presence of mental health problems, including psychological stress, depressive symptoms, bipolar affective disorder, and schizophrenia [88,89]. 

Opposite to the global trend, cardiovascular disease mortality rates in Europe show a trend of slow decline that varies across the region. These rates are higher in the Central (including Czechia) and Eastern European countries than in the Western European countries, highlighting the presence of an East–West health gap [90]. This gap grew during the 1970s and 1980s when CVD prevalence and mortality rates, as well as the impact of cardiovascular risk factors, declined in Western but not Eastern and Central Europe [91]. As a consequence of the socioeconomic and political changes of the 1990s, especially among the dissolution of the Soviet Union, this gap started to close in some Central and East European countries [90,91]. According to Global Burden Disease data [1], between 1990 and 2019, the age-adjusted rate of deaths by CVD per 100,000 inhabitants in Western European countries dropped by 24.2%, but in Central and Eastern Europe, it increased by 6.9% and 25.4%, respectively. During the same period, the rate of CVD deaths in Czechia dropped by 30.0%, above the average of the Western countries, reflecting an important reduction in the regional gap [1]. Improvements have also been seen in some other neighboring Eastern European countries, such as Poland (with 11.5% decrease), Slovakia (10.0% decrease), and Hungary (7.5% decrease). However, in Western European countries that share borders with Czechia, the advances have been more pronounced, for example, a 27.7% decrease in Austria and a 24.1% decrease in Germany [1]. With these dynamic changes in epidemiological metrics in mind, the question of cardiometabolic drivers and their impact becomes even more significant.

In line with increasing prevalence of diabetes in the Eastern, Central, and Southern Europe, the prevalence of T2D in Czechia is the sixth highest in EU [92]. However, the efforts of the healthcare system to reduce the present burden are evident. Around 83% of people with T2D in Czechia are aware of their condition (the European average is 59.3% [92]), 67% receive treatment, and the level of diabetes control is 43% [26]. The lack of success in effective prevention of obesity and diabetes demands a reinterpretation of how cardiometabolic drivers are expressed and interact among different populations. For more than 30 years, it has been understood that dysglycemia and adiposity drivers are mechanistically interconnected, but a translation of this understanding into an actionable model in early stages of risk has been elusive. The metabolic syndrome construct integrates five cardiometabolic components with a cut-off diagnosis of three risk factors. However, metabolic syndrome has not been recognized as a distinct disease, and, therefore, any value as a clinical diagnosis remains unclear [93]. To overcome these challenges, the CMBCD model was proposed to incorporate the natural history of the cardiometabolic disease in four stages: 1—risk; 2—pre-disease; 3—diseases; and 4—complications [4]. The CMBCD model combines the dysglycemia and abnormal adiposity drivers into the chronic care model as DBCD [6] and ABCD [5], with the presence of earlier primary drivers (genetics, environment, and behavior) and later metabolic drivers (hypertension and dyslipidemia) impacting the eventual cardiovascular disease phenotype [6]. The CMBCD model triggers proactive detection and stratification of subjects at risk using a culturally adapted approach to improve successful implementation and precision.

Shifting the main focus of the current healthcare framework to the primordial, primary prevention strategies and aggressive case finding of the disease at early stages could improve the longevity and quality of life of patients as well as reducing the burden on the healthcare system [4]. Concepts of primordial and primary prevention and health promotion only emerged in former Czechoslovakia after political change in 1990. Since then, a system of public health control is still being established. Various projects have already been successfully implemented; however, others require strategical improvement and support to effectively reduce the burden of dysglycemia, abnormal adiposity, and CVD [79]. Current strategies of secondary prevention could benefit from reinforcing the effective function of comprehensive cardiovascular rehabilitation multidisciplinary teams, interconnecting already existing tailored nutrition, physical activity, smoking cessation, and psychosocial interventions with pharmacological treatment of present metabolic abnormalities [94]. Intensive research is needed for constant updating of current prevention guidelines and ensuring their proper, homogeneous delivery to patients. For example, further clinical studies are essential to investigate whether restricting daily eating periods could mitigate circadian disruptions, promote weight loss, and improve cardiometabolic outcomes [95,96].

## 5. Conclusions

According to the findings of this literature search, an effective cardiometabolic preventive care program for Czechia needs to focus on shifting the traditional understanding and management of cardiometabolic factors towards a CMBCD complication-based model with particular attention given to the implementation of targeted interventions to avoid the progression of the cardiometabolic disease. In CMBCD stage 1, more attention should be given to primordial prevention with educational campaigns targeting unhealthy behaviors, incorporating community engagement, and reinforcing healthy eating habits; in stage 2, the implementation of primary prevention programs, e.g., the diabetes prevention program, promoting healthy dietary patterns, regular physical activity, and other lifestyle interventions should be transculturally adapted and scaled up; in stage 3, emphasis remains on the secondary prevention as a necessary intervention to mitigate disease progression and development of complications; in stage 4, tertiary prevention strategies to prevent the advance of complications and mortality are implemented with the highest healthcare cost compared to lifestyle preventive strategies.

## Figures and Tables

**Table 1 nutrients-13-02338-t001:** Important biological factors for cardiometabolic risk *.

Biological Factors	References
The FTO rs9939609 variant was associated with obesity FTO rs17817449 SNP was related to BMI in males and postmenopausal females *FTO* rs17817449 SNP was associated with the susceptibility to T2D and development of T2D complications The effect of the FTO rs17817449 variant on BMI is mediated through the effect on the basal metabolic rate, and its effect is more pronounced in women FTO and MC4R gene variants enhance the impact of an intensive lifestyle intervention on BMI decrease in overweight/obese children; this association was not confirmed in overweight females The TMEM18 rs7561317 was associated with underweight BDNF rs925946 and MC4R rs17782313 were associated with metabolic syndrome PCSK1 rs6235 was negatively related to increased blood glucose INSIG2 polymorphism has no significant effect on BMI and plasma lipids PPARα and PPARγ2 polymorphisms have no significant effect on anthropometric, biochemical, hormonal, and psychobehavioral characteristics of the subjects NYD-SP18 rs6971019 SNP is related to BMI in males; variants within NYD-SP18 and FTO genes revealed a significant additive effect on BMI values in males Prevalence of MC4R homozygous and heterozygous mutations among Czech obese children is 2.4%, and it is not associated with different responses to diet management	[7] [8] [9] [10] [11,12] [7] [7] [7] [13] [14] [15] [7,16]

* This is a partial list of research conducted on Czech populations regarding biological determinants of cardiometabolic risk. Abbreviations: ABCD—adiposity-based chronic disease, BDNF—brain-derived neurotrophic factor, BMI—body mass index, DBCD—dysglycemia-based chronic disease, FTO—fat mass and obesity-associated, INSIG2—insulin-induced gene 2, MC4R—melanocortin 4 receptor, NYD-SP18—testis development protein, PCSK1—proprotein convertase subtilisin/kexin Type 1, PPARs—peroxisome proliferator-activated receptors, SNP—single-nucleotide polymorphisms, TMEM18—transmembrane protein 18, T2D—type 2 diabetes.

**Table 2 nutrients-13-02338-t002:** Studies of prevalence of cardiometabolic risk factors in adults of Czechia.

Authors	Year of Publication (Data Collection Years)	Location	Participants (N)	Current Smoking (%)	Obesity (%)	Overweight (%)	Diabetes (%)	Prediabetes (%)	Hypertension (%)	Dyslipidemia (%)	Physical Inactivity (%)
Cifkova R et al. [17]	2020 (2016/2017)	Post-Monica National	1684 (25–64 y.o.)	M: 23.9 F: 20.9	M: 37.7 F: 27.6				M: 50.6 F: 33.5	M: 74.8 * F: 69.9 *	
Brož J. et al. [18]	2020 (2014)	EHES National	1189 (25–64 y.o.)	T: 29.5	T: 27.5	T: 36.4	T: 9.6 M: 11.5 F: 8.3	T: 27.8 M: 26.4 F: 28.7	T: 38.0	T: 62.0 ** T: 11.1 ***	
Movsisyan N et al. [19]	2017 (2013–2016)	Kardiovize population-based sample from Brno	2160 (25–64 y.o.)	T: 23.5 M: 25.3 F: 21.9	T: 19.0 M: 20.0 F: 18.1	T: 34.0 M: 43.0 F: 26.6	T: 5.1 M: 7.1 F: 3.5		T: 40.0 M: 46.0 F: 35.0	T: 68.8 ‖ M: 70.7 ‖ F: 67.1 ‖	T: 14.4 ¶ M: 16.8 ¶ F: 12.3 ¶
Čapková N et al. [20]	2017 (2014)	EHES National	1220 (25–64 y.o.)		M: 29.0 F: 25.0	M: 44.0 F: 30.0	M: 8.6 F: 5.7		M: 47.0 F: 26.0	M: 77.0 † F: 66.0 †	
Ministry of health report [21]	2014	National	-	T: 30.0	T: 17.0	T: 40.0	T: 7.0		M: 36 F: 31	M: 81.0 * F: 71.0 * Post Monica	T: 32.0 •
Cifkova R et al. [22]	2011 (2006–2009)	Post-Monica National	3612 (25–64 y.o.)	M: 31.9 F: 23.3	M: 32.4 F: 28.3	M: 44.4 F: 27.3	M: 9.4 F: 4.7		M: 47.8 F: 36.6	M: 81.0 ‡ F: 70.6 ‡	
Lustigova M et al. [23]	2018 (2002–2005)	HAPPIEE Havirov/Karvina, Hradec, Jihlava, Kromeriz, Liberec and Usti nad Labem	8499 (45–69 y.o.)	T: 26.0	T: 25.9		T: 11.8		T: 45.9		T: 55.5 ◦
Zejglicová K et al. [24]	2006 (1998–2002)	HELENA random 400 men and 400 women in 27 towns across Czechia	14190 (quest) 3669 (exam) 45–64 y.o.	T: 38.8	T: 22.5 M: 24.1 F: 21.2	T: 42.3 M: 50.4 F: 36.1			T: 41.4 M: 52.5 F: 32.9		T: 53.5 ◦ M: 52.2 ◦ F: 54.6 ◦
Cifkova R et al. [17]	2020 (1985)	Monica National	2570 (25–64 y.o.)	M: 45.0 F: 23.9	M: 19.7 F: 28.0		T: 6.1 M: 6.9 F: 5.4 [8]		M: 51.9 F: 42.5	M: 87.5 * F: 87.7 *	

* Dyslipidemia—total cholesterol ≥5.0 mmol/L; OR/AND (HDL-c < 1.0 mmol/L (men); HDL-c < 1.2 mmol/L (women)); OR/AND usage of lipid-lowering drugs. ** Dyslipidemia—total cholesterol ≥5 mmol/L OR/AND usage of lipid-lowering drugs. *** Dyslipidemia—HDL-c ≥ 1,2 mmol/L in men and ≥1mmol/L in women OR/AND usage of lipid-lowering drugs‖ Dyslipidemia—total cholesterol ≥ 5.0 mmol/L; OR/AND LDL-c ≥ 3 mmol/L; OR/AND triglycerides ≥1.7 mmol/L; OR/AND taking lipid-lowering drugs. † Dyslipidemia—total cholesterol >5.0 mmol/L; AND/OR HDL-c ≤ 1.2 mmol/L; AND/OR usage of lipid-lowering drugs. ‡ Dyslipidemia—total cholesterol ≥5.0 mmol/L; OR/AND (HDL-c < 1.0 mmol/L (men); HDL-c < 1.2 mmol/L (women)); OR/AND LDL-c ≥ 3.0 mmol/L; OR triglycerides ≥ 1.7 mmol/L; OR/AND usage of lipid-lowering drugs. ¶ Physical inactivity—achieved less than 600 MET-min/week. • Physical inactivity—achieved less than 150 min/week of moderate-intensity PA or/and 75 min/week vigorous-intensity PA. ◦ Physical inactivity—physical activity less than 3 h/week. Abbreviations: T—total; M—male; F—female.

**Table 3 nutrients-13-02338-t003:** Prevalence of adiposity- [25] and dysglycemia- [26] based chronic disease in a Czech population *.

**Adiposity-Based Chronic Disease (ABCD)**	**Total (%)**	**Male (%)**	**Female (%)**
No ABCD	37.2	33.2	40.6
Stage 0 ABCD (metabolically healthy)	2.3	4.0	0.9
Stage 1 ABCD (mild related complications)	31.4	28.4	33.9
Stage 2 ABCD (moderate–severe-related complications)	29.1	34.4	24.7
Total ABCD (abnormal adiposity)	62.8	66.8	59.4
**Dysglycemia-Based Chronic Disease (DBCD)**	**Total (%)**	**Male (%)**	**Female (%)**
No DBCD	30.6	30.9	30.3
Stage 1 DBCD (insulin resistance)	54.2	48.5	59.0
Stage 2 DBCD (prediabetes)	10.3	13.8	7.5
Stage 3 DBCD (type 2 diabetes)	3.7	4.9	2.6
Stage 4 DBCD (Type 2 diabetes and vascular complications)	1.2	1.9	0.6
Total DBCD (dysglycemia)	69.4	69.1	69.7

* In this study, only abnormal amounts of adiposity and distribution are detected (not function) for ABCD designation. Abbreviations: ABCD—adiposity-based chronic disease, CVD—cardiovascular disease, DBCD—dysglycemia-based chronic disease.

**Table 4 nutrients-13-02338-t004:** Health promotion in Czechia *.

Project Title	Main Themes	Setting
Healthy aging project [80]	Support of lifelong employment, learning, and social security of senior citizens Improvement in health and social services for seniors Awareness raising, anti-stigmatization, and anti-discrimination Housing and residential social services	Interdisciplinary cooperation, especially between health and social areas, including local governments, educational institutions, non-governmental organizations, and business. Cooperation with the National Network of Healthy Cities and reginal hygienic stations
Health Promoting School [81,82]	Improving the assortment of food in vending machines, school canteens, and buffets according to dietary recommendations Regular physical activity in school (providing place and time for physical activity every day) Educating teachers about healthy nutrition and sport and incorporating this knowledge into the whole educational process Full-time access to drinking water	Kindergartens and primary schools
Healthy City [83]	Increasing the number of parks and recreation zones in the city Improving public transport Improving the level of safety in the city Improving the cleanliness of public spaces Raising awareness about importance of sport and nutrition Raising awareness about non-communicable diseases (diabetes, CVD, oncological diseases) Tackling the obesogenic environment	Over 2152 cities and towns with 5423 million inhabitants (52% of Czech citizens)
Healthy Workplace [84]	Healthy and safe workplace conditions Coping with stress Decreasing the amount of work-related injuries Prevention of chronic musculoskeletal disorders and back pain Decreasing the level of noise pollution	Office and workspaces around the country
Delicious life [85]	Improving the knowledge of senior citizens about nutrition and motivation to achieve positive changes and be physical activity Improvement in social participation	Social care institutions
Educational program physical activity and nutrition [86]	Improving the quality of physical education classes in schools Adding physical activity breaks before and in between classes Creating afterschool clubs of various physical activities Providing all-day access to sugar-free drinks Increasing the amount of information about healthy nutrition in the curriculum Improving the quality and increasing the variety of food in school canteens Changing the assortment of the snacks in school vending machines in the direction of a healthier one	Primary Schools
The National Cycling Development Strategy of the Czech Republic for 2013–2020 [87]	Ensuring the financing of a cycling infrastructure Increasing the safety of bicycle traffic Methodological support for the development of bicycle transport in cities and the "Cycling Academy" project	Whole country

* This is a list of selected examples.

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
