# Peer review of "Dysglycemia and Abnormal Adiposity Drivers of Cardiometabolic-Based Chronic Disease in the Czech Population: Biological, Behavioral, and Cultural/Social Determinants of Health"

_nutrients, 2021, doi:10.3390/nu13072338_

Round 1
Reviewer 1 Report
Pavlovska et al. provide an overview on the prevalence of risk factors associated with dysglycemia and abnormal adiposity as the drivers of cardiometabolic-based chronic disease in the Czech population. This manuscript addresses an important and interesting topic, especially that the burden of metabolic disorders such as obesity and diabetes as well as cardiovascular disease is currently a subject of extensive epidemiological and clinical investigation. It is also worthy that the authors included in their analysis the new concepts associated with the cardiometabolic-based chronic disease (CMBD) models by Mechanick et al. Both the rationale and scientific content of the manuscript are valuable. However, I have several concerns that, in my opinion, deserve special attention in the revision.
I am not sure that the title of this review is adequate given the content of the manuscript. The title suggests that the review is focused on a comparison of an impact of biological vs. other risk factors on cardiometabolic risks in the Czech population. However, the manuscript is rather focused on the description of the prevalence of various groups of risk factors without any attempt of comparing a significance or mechanisms related to various risk factors. This issue requires attention
The abstract does not represent adequately and clearly the content of the manuscript. For example, there is no information in the abstract which risk factors or groups of factors are specifically considered in the review as dysglycemia and abnormal adiposity drivers. I also suggest to include in the abstract the description of methodology of this analysis and information that the CMBCD models were used. In addition, the authors are inconsistent in terms of an incidence of cardiovascular disease (CVD): is it increasing or decreasing (see L. 28 and L. 38)? These issues require further attention and clarification.
The Materials and Methods section needs improvements, especially in terms of the precise description of the methods that were used in the analysis of specific categories of risk factors. It would be desirable to add: (1) definitions of dysglycemia and abnormal adiposity which were used for the purpose of this analysis; (2) definitions of CMBCD, ABCD, and DBCD should be provided and explained in the beginning of the manuscript (e.g., “ABCD” is only just mentioned under Table 1 and in L. 133) because the concept of CMBCD models is not widely recognized; (3) detailed classification of groups of risk factors and lists of specific risk factors for CMBCD which were analyzed; (4) definitions of specific risk factors or groups of risk factors should be provided, e.g., biological risk factors, cardiometabolic risk factors (see L. 84), etc.; (5) because different types of sources of data were used for analyses of specific categories of risk factors, it should be explicitly indicated in the Methods section, e.g., clinical studies, epidemiological studies, WHO data, GBD data, etc.
The Results section and Discussion section include inconsistent sets of information. The Results section should describe the results of specific analyses while Discussion section should provide authors’ comments regarding the results and discuss these results with other studies/authors, etc. In this manuscript, in many places in the Results section, authors’ opinions and comments on the results are included (e.g., L. 85-88, L. 175-178, L. 199, L. 204, L. 215-217, L. 219, L. 239-241, L. 270-275, L. 291-294, L. 327-330), which should be included and extended in the Discussion section. This issue is important and requires further attention.
In the Results section, some other improvements are required. For example, there is no basic information on the number and types of studies/registries/surveys which were analyzed to obtain the results provided in the manuscript. This information should be provided in the beginning of Results section. In addition, it is difficult for reader to capture all important findings which are presented in the tables. So, I suggest to add a description of main findings, which are included in Table 1, Table2, and Table 3, in the text of Results section. Also, the definitions of “diabetes” (type 1 or type 2 or both were considered?), “hypertension”, “hypertension control”, or “dyslipidemia" should be provided.
In the Discussion section, the authors summarized pretty well main results of their analysis regarding the prevalence of various risk factors which are drivers for CMBCD, ABCD, and DBCD. However, I suggest to extend this section by more detailed description of clinical and epidemiological implications associated with high prevalence of cardiometabolic risk factors in the Czech population. Also, more discussion in terms of comparisons with other populations would be needed. It would be also desirable to add a paragraph in section 4 about potential therapeutic approaches targeting cardiometabolic risks for reducing the burden of CMBDC and CVD. In addition, it would be useful to indicate current recommendations (relevant references) regarding primary and secondary prevention for CMBCD and CVD. Also, it is important to further improve section 4 through specifying future directions of basic and clinical research on the topics related to new preventive and therapeutic strategies for cardiometabolic risks, e.g., targeting circadian rhythm disruption. I suggest adding a paragraph in section 4 devoted to this topic and including relevant references. Example relevant references that would fit in this context are:
Ambrosetti M, Abreu A, Corrà U, et al. Secondary prevention through comprehensive cardiovascular rehabilitation: From knowledge to implementation. 2020 update. A position paper from the Secondary Prevention and Rehabilitation Section of the European Association of Preventive Cardiology. Eur J Prev Cardiol. 2020 Mar 30:2047487320913379. doi: 10.1177/2047487320913379.
Świątkiewicz, I., A. Woźniak, P.R. Taub. Time-Restricted Eating and Metabolic Syndrome: Current Status and Future Perspectives. Nutrients, 13, 221; doi.org/10.3390/nu13010221, 2021.
de Cabo R, Mattson MP. Effects of Intermittent Fasting on Health, Aging, and Disease. N Engl J Med. 2019 Dec 26;381(26):2541-2551. doi: 10.1056/NEJMra1905136.
The scientific writing requires further attention because some statements can be made clearer to provide better understanding of specific message or content, especially in the Results and Discussion sections. The language and style also require checking and minor improvements.
Author Response
Response to Reviewer 1 Comments
Point 1: I am not sure that the title of this review is adequate given the content of the manuscript. The title suggests that the review is focused on a comparison of an impact of biological vs. other risk factors on cardiometabolic risks in the Czech population. However, the manuscript is rather focused on the description of the prevalence of various groups of risk factors without any attempt of comparing a significance or mechanisms related to various risk factors. This issue requires attention
Response 1: The title has been corrected
Point 2: The abstract does not represent adequately and clearly the content of the manuscript. For example, there is no information in the abstract which risk factors or groups of factors are specifically considered in the review as dysglycemia and abnormal adiposity drivers. I also suggest to include in the abstract the description of methodology of this analysis and information that the CMBCD models were used. In addition, the authors are inconsistent in terms of an incidence of cardiovascular disease (CVD): is it increasing or decreasing (see L. 28 and L. 38)? These issues require further attention and clarification.
Response 2: The abstract has been modified to reflect the content of the review better
Point 3: The Materials and Methods section needs improvements, especially in terms of the precise description of the methods that were used in the analysis of specific categories of risk factors. It would be desirable to add: (1) definitions of dysglycemia and abnormal adiposity which were used for the purpose of this analysis; (2) definitions of CMBCD, ABCD, and DBCD should be provided and explained in the beginning of the manuscript (e.g., “ABCD” is only just mentioned under Table 1 and in L. 133) because the concept of CMBCD models is not widely recognized; (3) detailed classification of groups of risk factors and lists of specific risk factors for CMBCD which were analyzed; (4) definitions of specific risk factors or groups of risk factors should be provided, e.g., biological risk factors, cardiometabolic risk factors (see L. 84), etc.; (5) because different types of sources of data were used for analyses of specific categories of risk factors, it should be explicitly indicated in the Methods section, e.g., clinical studies, epidemiological studies, WHO data, GBD data, etc.
Response 3: 1,2) Added the sub-section 2.1 Variables definition to the Materials and methods section. 3,4) In the first sentence of the Materials and Methods section the classification of risk factors was provided. Additional, the more detailed algorithm of a literature search of studies regarding all of the drivers has been described 5) Added the description of other sources at the end of the Materials and Methods section
Point 4: The Results section and Discussion section include inconsistent sets of information. The Results section should describe the results of specific analyses while Discussion section should provide authors’ comments regarding the results and discuss these results with other studies/authors, etc. In this manuscript, in many places in the Results section, authors’ opinions and comments on the results are included (e.g., L. 85-88, L. 175-178, L. 199, L. 204, L. 215-217, L. 219, L. 239-241, L. 270-275, L. 291-294, L. 327-330), which should be included and extended in the Discussion section. This issue is important and requires further attention.
Response 4: The Results section has been cleared from the parts that belong to the Discussion section
Point 5: In the Results section, some other improvements are required. For example, there is no basic information on the number and types of studies/registries/surveys which were analyzed to obtain the results provided in the manuscript. This information should be provided in the beginning of Results section.
Response 5: The basic information on the number of studies has been added to the beginning of the Result section.
Point 6: it is difficult for reader to capture all important findings which are presented in the tables. So, I suggest to add a description of main findings, which are included in Table 1, Table2, and Table 3, in the text of Results section. Also, the definitions of “diabetes” (type 1 or type 2 or both were considered?), “hypertension”, “hypertension control”, or “dyslipidemia" should be provided.
Response 6: Added the basic findings from the tables to the text. Added the definitions of hypertension, hypertension control in the Variables Definition sub-section of the Materials and Methods section.
Point 7: I suggest to extend this section by more detailed description of clinical and epidemiological implications associated with high prevalence of cardiometabolic risk factors in the Czech population. Also, more discussion in terms of comparisons with other populations would be needed.
Response 7: Epidemiological (comparison with other countries) and clinical (diabetes awareness and control) implications are added in the Discussion. Epidemiological implications – paragraph 2 of the discussion, clinical implication – paragraph 3 of the Discussion
Point 8: It would be also desirable to add a paragraph in section 4 about potential therapeutic approaches targeting cardiometabolic risks for reducing the burden of CMBDC and CVD. In addition, it would be useful to indicate current recommendations (relevant references) regarding primary and secondary prevention for CMBCD and CVD. Also, it is important to further improve section 4 through specifying future directions of basic and clinical research on the topics related to new preventive and therapeutic strategies for cardiometabolic risks, e.g., targeting circadian rhythm disruption. I suggest adding a paragraph in section 4 devoted to this topic and including relevant references. Example relevant references that would fit in this context are:
Ambrosetti M, Abreu A, Corrà U, et al. Secondary prevention through comprehensive cardiovascular rehabilitation: From knowledge to implementation. 2020 update. A position paper from the Secondary Prevention and Rehabilitation Section of the European Association of Preventive Cardiology. Eur J Prev Cardiol. 2020 Mar 30:2047487320913379. doi: 10.1177/2047487320913379.
Świątkiewicz, I., A. Woźniak, P.R. Taub. Time-Restricted Eating and Metabolic Syndrome: Current Status and Future Perspectives. Nutrients, 13, 221; doi.org/10.3390/nu13010221, 2021.
de Cabo R, Mattson MP. Effects of Intermittent Fasting on Health, Aging, and Disease. N Engl J Med. 2019 Dec 26;381(26):2541-2551. doi: 10.1056/NEJMra1905136.
Response 8: Added entirely new paragraph 4 to the discussion, which briefly explains the current recommendations and some future direction for improvement. Mentioned references were also used.
Point 9: The scientific writing requires further attention because some statements can be made clearer to provide better understanding of specific message or content, especially in the Results and Discussion sections. The language and style also require checking and minor improvements.
Response 9: Scientific writing has been improved
Reviewer 2 Report
This is a rather well written review article that focuses on the changing trends of cardiometabolic disease risk factors in the Czech Republic. The review covers a wide range of health determinants. The discussion focusses strictly on studies conducted in the Czech Republic, and where available, relates data from the Czech Republic to that of Central or Eastern Europe. As such, I believe that this work has value to the Czech public health community. It does not merely read as though trends that began in the US/UK decades ago are finally catching up to the Czech Republic. I have a few comments to help improve the manuscript.
Major Comments:
- Is there any data available for the percentage of patients on statins or other lipid lowering drugs or anti-hypertensive medications that could help explain the trend for reduction in cardiovascular disease mortality despite increasing adiposity and decreasing physical activity?
- Line 196: “Since 1989, the quality of diet has improved, with cereals, pulses, vegetables and fruits becoming more popular, and consumption of milk, meat, animal fat, refined sugar and starch decreasing [22]”. Could this be described in greater detail and possibly quantified? If true for Chechia, these trends would go rather against those observed in much of the rest of the world. They also generally do not track with increases in adiposity and dysglycemia and would be reflective of an improving health literacy.
- Is there any data available for trends in processed food consumption in Chechia relative to Central/Eastern Europe?
Minor Comments:
- I suggest reformatting Table 1 so that the references align with the biological factor that they are referring to. I also think left-aligning the text and using bullet points would enhance readability.
- Also Table 4 is difficult to read, and it is difficult to decipher where the themes from one study end and another begin. Again, I would suggest left-aligning the text and using bullet points to distinguish the themes.
Author Response
Response to Reviewer 2 Comments
Point 1: Is there any data available for the percentage of patients on statins or other lipid-lowering drugs or anti-hypertensive medications that could help explain the trend for reduction in cardiovascular disease mortality despite increasing adiposity and decreasing physical activity?
Response 1: Added the percentage of patients on lipid-lowering medications, antihypertensive medications to the second paragraph of the result sub-section 3.1. Epidemiology of Cardiometabolic risk factors. One sentence regarding this was also added to abstract
Point 2: Line 196: “Since 1989, the quality of diet has improved, with cereals, pulses, vegetables and fruits becoming more popular, and consumption of milk, meat, animal fat, refined sugar and starch decreasing [22]”. Could this be described in greater detail and possibly quantified? If true for Chechia, these trends would go rather against those observed in much of the rest of the world. They also generally do not track with increases in adiposity and dysglycemia and would be reflective of an improving health literacy.
Response 2: Added more information on the trend to the second paragraph of sub-section 3.3 Eating Habits
Point 3: Is there any data available for trends in processed food consumption in Chechia relative to Central/Eastern Europe?
Response 3: Added the relevant information to the first paragraph of sub-section 3.3 Eating Habits
Point 4: I suggest reformatting Table 1 so that the references align with the biological factor that they are referring to. I also think left-aligning the text and using bullet points would enhance readability.
Response 4: Table 1 has been reformatted
Point 5: Also Table 4 is difficult to read, and it is difficult to decipher where the themes from one study end and another begin. Again, I would suggest left-aligning the text and using bullet points to distinguish the themes.
Response 5: Table 4 has been reformatted
Reviewer 3 Report
Dear Authors, the comments are provided in the manuscript.

Author Response
Response to Reviewer 3 Comments
Point 1: In my opinion, the manuscript is rather a description of the health situation of the Czech population and health care. It is difficult to see any deeper meaning. The manuscript seems chaotic. It lacks specific conclusions that could be used in other populations or that would guide future research
Response 1: The structure of the manuscript has been improved (the author’s comments on the Results were moved to the Discussion section). Clinical and epidemiological implications of the findings have been added to the Discussion section. The conclusion has also been improved, with information on proposed interventions to reduce the burden of CMBCD and CVD that could be adopted in other populations.
Point 2: Group size (in Materials and Methods section)
Response 2: Group size was not one of the inclusion criteria.
Point 3: Which number does concern male and female (third paragraph of Epidemiology of Cardiometabolic Risk Factors sub-section)
Response 3: Clarification has been added.
Point 4: Please, explain in footnotes what is T, F, M (Table 2)
Response 4: Explanation added.
Point 5: Correct the font (3.2 Socio-Demographic Characteristics)
Response 5: Font has been corrected.
Point 6: In my opinion, this is not a Discussion, but a Summary of the results
Response 6: The flow of the Discussion has been improved (added epidemiological and clinical implications of the findings), separate paragraph for summarizing current and potential interventions for reducing the burden of CMBCD and CVD.
Point 7: Please, explain what kind of intervention (Conclusion)
Response 7: conclusion has been improved, with a more detailed description of proposed interventions.
Round 2
Reviewer 1 Report
The authors adequately addressed my comments.
Reviewer 2 Report
The authors have sufficiently addressed my comments and the quality of the manuscript is improved. A few typos noted:
Line 383: Change Easter European to Eastern European.
Line 392: Change advances to advanced.
Reviewer 3 Report
Dear Authors, thank you for making changes. The current version of the manuscript has a much better flow, and thanks to the overall content modification it has gained a higher value.